# Neoindustrialization—Reflections on a New Paradigmatic Approach for the Industry: A Scoping Review on Industry 5.0

**Ricardo Pereira *** and **Neri dos Santos**

Knowledge Engineering Department, Federal University of Santa Catarina, Florianópolis 88040-900, Brazil; nerisantos@gmail.com
* Correspondence: rikardop@gmail.com

**Abstract:** *Background*: The Industry 5.0 emerges as a new paradigm for the industry by considering sustainability, human-centered approaches, organizational resilience, and interaction between humans and machines as its core values. This new trend for the future of the industry is referred to as neoindustrialization. Due to being a topic in development, there is still no precise consensus on its definition, which prompted the current study to comprehensively investigate and analyze the existing literature on Industry 5.0. *Methods*: The method employed was a scoping review, examining publications from various databases and academic journals, including those specific to the Brazilian context. *Results*: The results indicate a transition towards an industry that meets societal demands and respects planetary boundaries, aspects that were overlooked by Industry 4.0. *Conclusions*: In this new scenario, the industry reassumes its leadership by combining technology with new strategies and organizational models. Furthermore, it undergoes organizational changes to align its structure, operations, human resources, and new practices, aiming to meet the demands of society and all stakeholders involved. To achieve this, it is necessary to create an environment conducive to innovation and entrepreneurship, promoting the development of qualified human capital, investments in research and development, and strengthening partnerships between the public and private sectors. A successful neoindustrialization policy will generate high-quality jobs and foster economic growth. Industry 5.0 is the paradigm that will prevail in the 21st century. It is not a matter of speculation; it is an inseparable and inevitable reality. Otherwise, the industry will be relegated to a secondary role in the process of digital and social transformation.

**Keywords:** Industry 5.0; scoping review; neoindustrialization; the fifth industrial revolution

## 1. Introduction

Industry 4.0 has become the standard in the industry in recent years, with its technologies being implemented and positively impacting all sectors. However, these developments still fall short of achieving the desired results, neglecting the environment and prioritizing machines over humans [1–3].

This context has prompted research and discussions that have been undertaken since 2017. In 2021, the European Commission formalized the call for a new Industrial Revolution with the publication of a document titled "Industry 5.0: Towards a Sustainable, Human-centric, and Resilient European Industry". This study resulted from two workshops held in 2020, which discussed the need to recognize the power of the industry in achieving social objectives, as well as jobs and growth, while considering the well-being of industrial workers, sustainability, resilience, and other issues related to productivity and efficiency in the supply chain [1,4,5].

Industry 5.0, referred to as Neoindustrialization in this study, emerges as an alternative to a set of challenges in the contemporary world (climate change, rapid consumption of non-renewable resources and energy, environmental pollution, and social injustice, among others) that have been amplified by the COVID-19 pandemic and the Russia-Ukraine

conflict, elevating the complexity and dynamics of this context to levels not seen since World War II (1939–1945).

This panorama provides sufficient evidence that current practices need to be changed, pointing towards a new reality in the industry represented by a new paradigmatic approach, where human values, the environment, and the planet are preserved and respected [1,2,4].

Due to being a topic in development, there is still no precise consensus on its definition [6]. To address the need for a better understanding of the concept, this study aims to investigate and summarize the current state of knowledge about Industry 5.0. It does so through a scoping review conducted on the Scopus, Web of Science, Scielo, and Spell databases, as well as the main business and management journals in Brazil and the proceedings of EnANPAD, the largest business and management congress in Latin America. This approach portrays the Brazilian context and identifies the current stage of research development that considers Industry 5.0 as the object of study.

The article is structured into four sections, including this introduction. The following section addresses the research methodology procedures. The Section 3 presents the results derived from the literature survey conducted in the review. The Section 5 concludes the present study, providing final considerations, limitations, and recommendations for future work.

## 2. Materials and Methods

This study showcases the research conducted on Industry 5.0, utilizing the scoping review method [7–9]. This methodological approach aims to explore the key concepts of a topic, investigate the dimension, scope, and nature of the study, condense and publish the data, and identify existing research gaps [7]. Moreover, scoping reviews are useful for examining emerging evidence when it is still unclear what other more specific questions can be posed for synthesis [9].

According to Munn et al. [10], scoping reviews are suggested in several instances. They can serve as a preliminary step to a systematic review, helping to identify the available evidence in a particular field. Scoping reviews are also useful for identifying and analyzing knowledge gaps, shedding light on key concepts and definitions found in the literature. Furthermore, they offer insight into how research is conducted on a specific topic or field and can help identify key characteristics or factors associated with a particular concept.

In summary, scoping reviews are suitable in cases where a topic or body of knowledge is new, under construction, or poorly addressed in the literature. This method is highly recommended in the present case, which requires further discussion on the definition of the term Industry 5.0, investigating its theoretical and practical impact, as well as its implications for society.

This study considered the guidelines proposed by [7] and expanded upon by [8]. The methodological procedures used are described below:

Stage (1) Identification of the research question and definition of the study objective: This stage aims to align, clarify, and link the objective to the research question. The research question in a scoping review should be clearly defined as it plays a significant role in the subsequent stages [7–9]. The research question for this study is: What is the current state of research in the field of Industry 5.0? The objective, in turn, is to investigate and summarize the current state of research in the field of Industry 5.0.

Stage (2) Identification of relevant studies: In this stage, the identification of relevant studies and the plan for searching, including search terms, sources of research, time frame, and language, are established. The aim is to balance feasibility with the scope's breadth and coverage. It is important to develop and align inclusion criteria with the objective and research question [7–9].

For this study, the search strategy was defined using the string TITLE ("industry 5.0") AND (LIMIT-TO (DOCTYPE, "ar") OR LIMIT-TO (DOCTYPE, "re")) in the Scopus, Web of Science, Scielo, and Spell databases. There was no temporal delimitation. To portray the Brazilian context and consequently identify the current stage of research development

considering Industry 5.0 as the object of study, a search was conducted in the main business and management journals in Brazil, as well as the proceedings of ENANPAD (Meeting of the Brazilian Association of Postgraduate and Research in Business Administration), the largest scientific congress in management and business in Latin America. The search was restricted to articles and reviews in English and Portuguese. The number of publications obtained in this stage is presented in Figure 1.

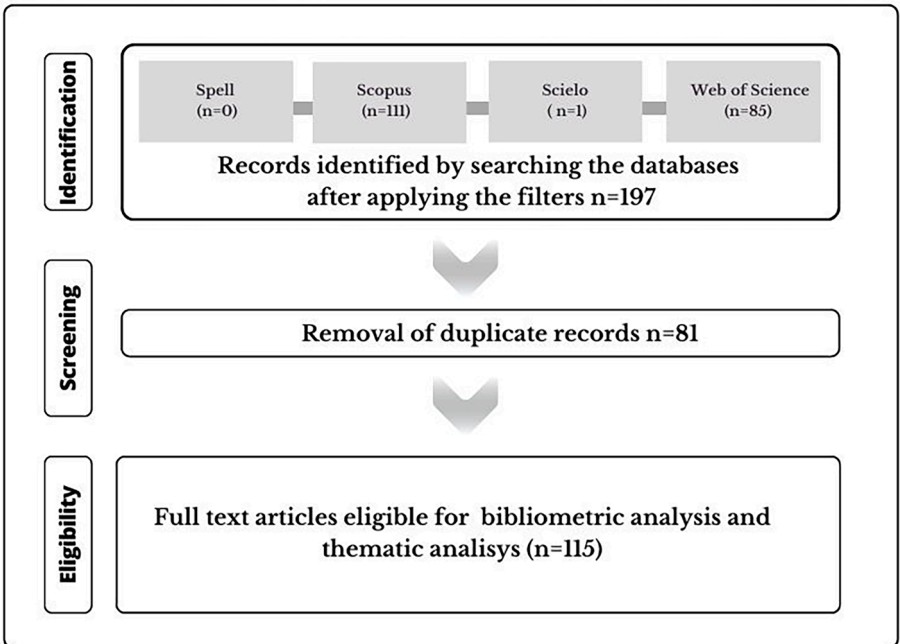

**Figure 1.** Methodological Procedures.

Stage (3) Selection of studies: The process of selecting publications for analysis is not a linear procedure but rather an iterative one that involves literature search, refining the search strategy, and reviewing articles for inclusion in the study. Here, we describe the planned approach for evidence searching, selection, data extraction, and presentation of evidence [7–9]. In this stage, the abstracts, keywords, and titles of the publications were read, organized, and identified to determine the selected studies. The number of publications obtained in this stage is presented in Figure 1.

Stage (4) Data Mapping: An "analytical descriptive" method can be adopted to extract contextual or process-oriented information from each study [7–9]. After selecting the articles most suitable for the research objective, the key data from these studies were extracted and indexed in a synthesis matrix [11].

Stage (5) Clustering, summarizing, and reporting of results: Implications of the study's findings for policies, practices, or research are identified. Selecting the evidence [7–9]. In this stage, the selected publications from the previous stages were analyzed through thematic analysis using Braun and Clarke's approach [12] and visualization of similarities using the Vosviewer® tool. The use of these techniques allowed for the identification, coding, and grouping of themes during the process of reading and analyzing the articles. The data collected in this final stage will be presented in the following section.

## 3. Results

*Industry 5.0 in Numbers*

The data mapping process carried out during Stage 4 of the methodological procedures in this study enabled the realization of bibliometric analysis. The techniques of performance analysis and scientific mapping were employed, enabling the assessment of authors, organizations, and countries based on their publications and citations. The relative contribution of each actor to the field of knowledge is evaluated and used to identify

the most productive and impactful ones. Subsequently, scientific mapping is conducted, which analyzes the structure and dynamics of knowledge production through a graphical representation of relationships among authors, topics, and institutions, among others. This technique has gained prominence in bibliometric analysis, largely due to the development of software/tools that facilitate mapping, making it an important methodological option for representing and analyzing the various networks that form within scientific knowledge production.

The literature search that supported the bibliometric analysis was conducted in the scientific databases Scopus, Scielo, Spell, and Web of Science. These databases were chosen considering their broad coverage, interdisciplinary nature, and the level of structure and data exportability they provide. The set of publications depicted in the graph in Figure 2 was identified. There is a noticeable sharp increase in publications, particularly in the last year, which had approximately five times more publications than in 2021. These data indicate a growing interest in studies/research related to Industry 5.0.

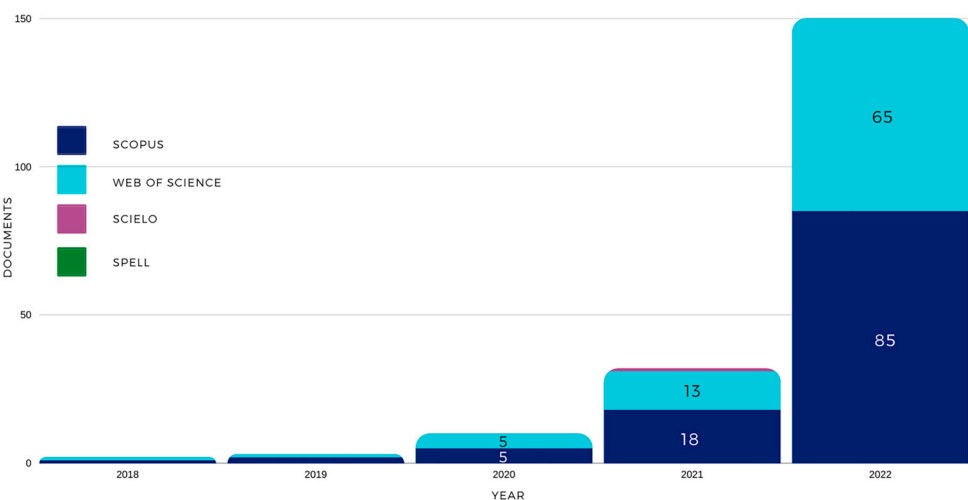

**Figure 2.** Number of publications on Industry 5.0 over time.

On the other hand, when comparing the number of research conducted on Industry 4.0 with that of Industry 5.0, there is a significant difference between the two paradigms, with Industry 4.0 having a much larger number of publications (Figure 3).

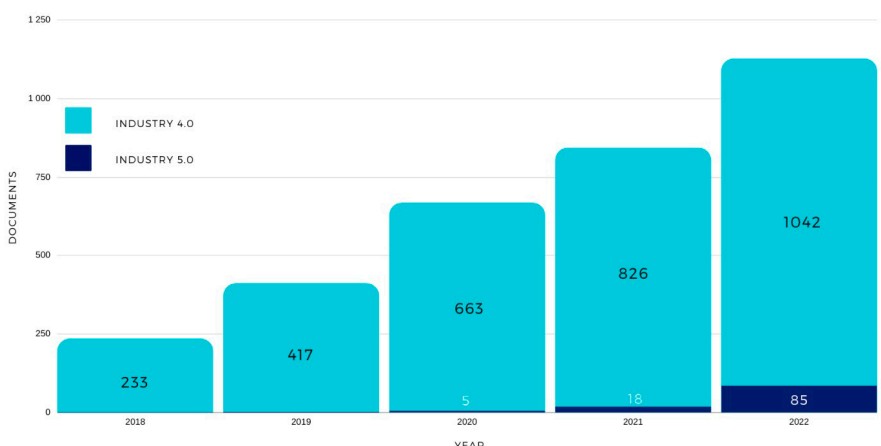

**Figure 3.** Number of publications on Industry 4.0 and Industry 5.0 over time.

Through this analysis, it is evident that studies on Industry 4.0 continue to attract academic interest, and both paradigms coexist.

Table 1 lists the top twenty publications on Industry 5.0 with the highest number of citations. The first article on the list is authored by [13] and titled "Industry 5.0—A human-centric solution", which has been cited 491 times. This article is conceptual in nature, where the author describes Industry 5.0 as a situation "where robots are intertwined with the human brain and work as collaborators rather than competitors" (p. 1). According to the author, this closer collaboration between robots and humans will be achieved through brain-machine interfaces and advancements in artificial intelligence. From this perspective, Industry 5.0 can help increase productivity without removing human workers from the industry. Additionally, the article discusses the potential implications of Industry 5.0 on the economy and productivity, and how the collaboration between robots and humans can be a solution to some of the challenges faced by the industry. The article also delves into the discussion of concerns and considerations that need to be taken into account when adopting Industry 5.0.

The second article in the table, "Industry 5.0: A survey on enabling technologies and potential applications", authored by [14], discusses the technologies that enable the new paradigm for the industry. It provides an overview of potential applications and supporting technologies, such as smart healthcare, cloud manufacturing, and supply chain management. The article also highlights several technologies, including edge computing, digital twins, collaborative robots, the Internet of Things, blockchain, and 6G networks.

The article "Birth of Industry 5.0: Making Sense of big data with artificial intelligence, 'the Internet of Things' and next-generation technology policy" by [15], with 326 citations, discusses the impacts of extreme automation, the Internet of Things (IoT), artificial intelligence (AI), and Industry 4.0 on the implementation of big data sets. The authors argue that while extreme automation has the potential to increase efficiency and productivity, it also presents significant vulnerabilities, including systemic risks and the creation of new social and political power structures. To overcome these issues, the article proposes Industry 5.0, which aims to democratize the co-production of knowledge from big data sets, based on the new concept of symmetrical innovation. Hence, Industry 5.0 holds the promise of securely leveraging extensive automation and big data by employing innovative technology policies and responsible implementation practices

The next article on the list, by [5], titled "Industry 4.0 and Industry 5.0—Inception, conception and perception", states that many countries have introduced strategic initiatives similar to Industry 4.0, and a considerable amount of research effort has been dedicated to the development and implementation of some Industry 4.0-related technologies. According to the authors, Industry 5.0 is a more values-oriented approach, focusing on issues such as sustainability, inclusivity, and collaboration. The article also explores five questions that arise from the coexistence of Industry 4.0 and 5.0, aiming to stimulate debate and discussion around these topics.

Longo et al. [2], on the other hand, follow the same line as [13], addressing the relationship between humans and machines, indicating the advent of the "augmentation era" where both humans and robots collaborate and work in symbiosis. This article examines the impact of technology on workers and society as a whole.

These first five listed articles present different perspectives regarding the characterization of this new phenomenon called Industry 5.0. A closer analysis of the Table 1 list may indicate some research trends on Industry 5.0, such as:

(a) Theoretical/conceptual studies or systematic literature reviews/bibliometric analyses that aim to explain/detail/expand the discussion on Industry 5.0 and related topics (I1, I4, I8, I13, I17).
(b) Studies that address the enabling technologies of Industry 5.0 (I2, I3, I6, I7, I12).
(c) Studies that relate Industry 5.0 to its core values, such as sustainability (I12, I19), human-centricity (I1, I2, I3, I5, I10, I18, I20).
(d) Studies that analyze the application of Industry 5.0-related technologies in specific/concrete contexts/cases (I9, I15, I16).

**Table 1.** Most cited articles on Industry 5.0 *.

| Id | Year | Author | Title/Source | Citations |
|---|---|---|---|---|
| I1 | 2019 | Nahavandi, S. [13] | Industry 5.0-a human-centric solution | 491 |
| I2 | 2022 | Maddikunta, P.K.R., Pham, Q.-V., B, P., (. . .), Ruby, R., Liyanage, M. [14] | Industry 5.0: A survey on enabling technologies and potential applications | 347 |
| I3 | 2018 | Özdemir, V., Hekim, N. [15] | Birth of Industry 5.0: Making Sense of Big Data with Artificial Intelligence, "the Internet of Things" and Next-Generation Technology Policy | 326 |
| I4 | 2021 | Xu, X., Lu, Y., Vogel-Heuser, B., Wang, L. [5] | Industry 4.0 and Industry 5.0—Inception, conception and perception | 292 |
| I5 | 2020 | Longo, F., Padovano, A., Umbrello, S.. [2] | Value-oriented and ethical technology engineering in industry 5.0: A human-centric perspective for the design of the factory of the future | 181 |
| I6 | 2020 | Javaid, M., Haleem, A., Singh, R.P., (. . .), Raina, A., Suman, R. [16] | Industry 5.0: Potential applications in COVID-19 | 119 |
| I7 | 2020 | Aslam, F., Aimin, W., Li, M., Rehman, K.U. [17] | Innovation in the era of IoT and industry 5.0: Absolute innovation management (AIM) framework | 106 |
| I8 | 2020 | Javaid, M., Haleem, A. [18] | Critical components of industry 5.0 towards a successful adoption in the field of manufacturing | 103 |
| I9 | 2021 | ElFar, O.A., Chang, C.-K., Leong, H.Y., (. . .), Chew, K.W., Show, P.L. [19] | Prospects of Industry 5.0 in algae: Customization of production and new advance technology for clean bioenergy generation | 89 |
| I10 | 2022 | Lu, YQ; Zheng, H; (. . .); Bao, JS [20] | Outlook on human-centric manufacturing towards Industry 5.0 | 64 |
| I11 | 2022 | Gürdür Broo, D., Kaynak, O., Sait, S.M. [21] | Rethinking engineering education at the age of industry 5.0 | 58 |
| I12 | 2021 | Fraga-Lamas, P., Lopes, S.I., Fernández-Caramés, T.M. [22] | Green iot and edge AI as key technological enablers for a sustainable digital transition towards a smart circular economy: An industry 5.0 use case | 57 |
| I13 | 2022 | Akundi, A., Euresti, D., Luna, S., (. . .), Lopes, A., Edinbarough, I. [23] | State of Industry 5.0—Analysis and Identification of Current Research Trend | 46 |
| I14 | 2022 | Carayannis, E.G., Morawska-Jancelewicz, J. [24] | The Futures of Europe: Society 5.0 and Industry 5.0 as Driving Forces of Future Universities | 42 |
| I15 | 2021 | Carayannis, E.G., Draper, J., Bhaneja, B. [25] | Towards Fusion Energy in the Industry 5.0 and Society 5.0 Context: Call for a Global Commission for Urgent Action on Fusion Energy | 40 |
| I16 | 2022 | Khan, W.U., Ihsan, A., Nguyen, T.N., Ali, Z., Javed, M.A. [26] | NOMA-Enabled Backscatter Communications for Green Transportation in Automotive-Industry 5.0 | 38 |
| I17 | 2021 | Madsen, D.Ø., Berg, T. [6] | An exploratory bibliometric analysis of the birth and emergence of industry 5.0 | 35 |
| I18 | 2022 | Sindhwani, R., Afridi, S., Kumar, A., (. . .), Luthra, S., Singh, P.L. [1] | Can industry 5.0 revolutionize the wave of resilience and social value creation? A multi-criteria framework to analyze enablers | 34 |
| I19 | 2022 | Yin, S., Yu, Y. [27] | An adoption-implementation framework of digital green knowledge to improve the performance of digital green innovation practices for industry 5.0 | 31 |
| I20 | 2022 | Saniuk, S., Grabowska, S., Straka, M [28] | Identification of Social and Economic Expectations: Contextual Reasons for the Transformation Process of Industry 4.0 into the Industry 5.0 Concept | 30 |

* Data extraction date: 19 February 2023.

These observed trends in the most cited articles are reflected and substantiated by the keyword co-occurrence map depicted in Figure 4. This co-occurrence technique enables the identification of relationships among the keywords within the selected set of documents, facilitating the conceptual mapping of a specific knowledge domain. When keywords co-occur frequently in the documents, it indicates that the concepts associated with them are related. In this analysis, the content of the documents is used to construct a measure of similarity. The outcome of the analysis is a network of interconnected themes that collectively represent the conceptual landscape of a particular field [29].

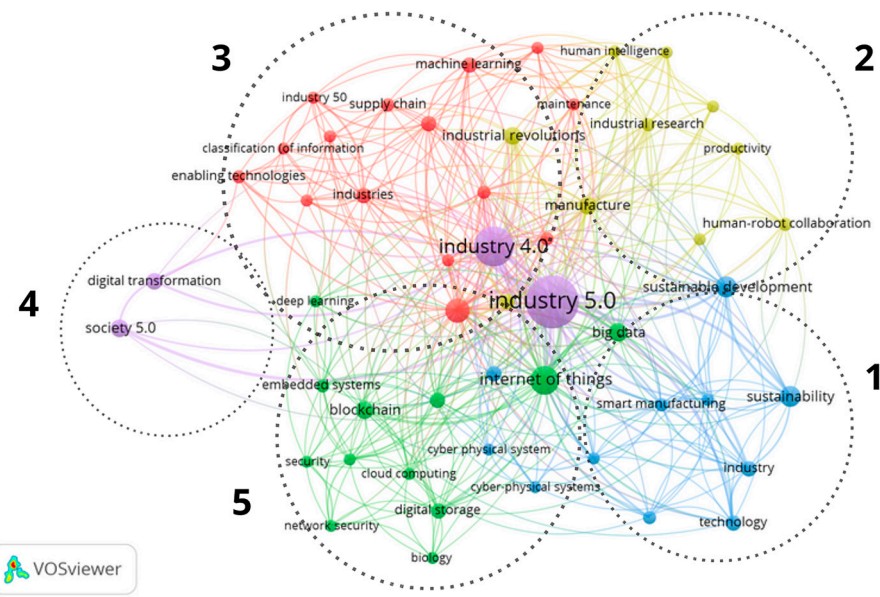

**Figure 4.** Keyword co-occurrence map.

The graph presented in Figure 4 represents the analysis of keyword co-occurrence among the articles in the bibliographic portfolio of analysis. The network is divided into five clusters represented by the colors blue, yellow, red, purple, and green.

Each cluster aggregates keywords that exhibit similarities and interactions among related themes. Cluster 1, represented by the color blue, includes keywords related to studies that link Industry 5.0 to one of its dimensions: sustainability. Cluster 2, represented by the color yellow, shows the interaction between keywords referring to Industry 5.0 and the collaboration between humans and robots (which gives rise to the term co-bot). Cluster 3, represented by the color red, groups keywords related to the technologies that enable Industry 5.0. Cluster 4, represented by the color purple, gathers keywords associated with Industry 5.0, digital transformation, and the societal impact of this new industry paradigm (Society 5.0). Finally, Cluster 5, represented by the color green, aggregates keywords related to Industry 5.0 technologies but from the perspective of cybersecurity and organizational resilience.

Figure 5 displays the citation map and its variations regarding documents (quadrant 1), countries (quadrant 2), research institutions (quadrant 3), and sources (quadrant 4), constructed using the VOSviewer® tool (Copyright © 2023 Centre for Science and Technology Studies, Leiden University, Leiden, The Netherlands). The maps consist of nodes that, when clustered together, generate clusters. The larger and more centralized a particular node is, the greater its representativeness within the set of publications.

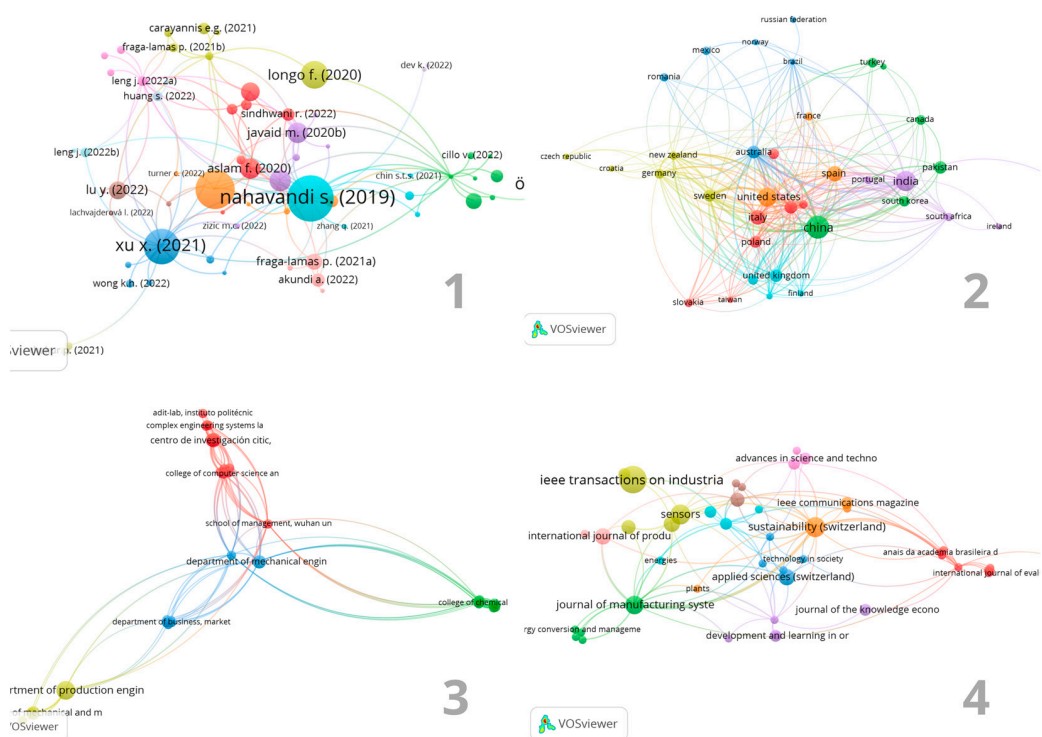

**Figure 5.** Bibliometric analyses related to Industry 5.0.

In the citation analysis, related to quadrant 1, documents that received at least 1 citation from other documents in the portfolio of publications under bibliometric analysis were considered. The analysis considered 64 items and generated 15 clusters and 123 links. This analysis explicitly highlights and corroborates the information presented in Table 1, particularly regarding the most cited articles: [13], represented by a light blue cluster; [14], represented by an orange cluster; [5], represented by a dark blue cluster; [15], represented by a green cluster; and [2], represented by a yellow cluster, are the main documents in this analysis.

Quadrant 2 of Figure 5 enables the analysis of citations by countries, highlighting China (green cluster), India (purple cluster), the United States (orange cluster), and Sweden, Germany, and New Zealand (yellow cluster) as the main contributors. This information indicates that these countries are home to institutions/researchers with the highest number of citations in the selected publications for analysis, demonstrating their prominence in research on Industry 5.0.

Quadrant 3 presents the citation map related to research institutions and is closely related to quadrant 2, as the countries mentioned there house the institutions with the greatest prominence in studies on Industry 5.0. Prominent institutions in the analysis include the Royal Institute of Technology-KTH (Sweden) and the University of Auckland (New Zealand) in the yellow cluster, and Jamia Millia Islamia University (India) in the blue cluster, as well as Wuhan University of Technology (China), among others.

Quadrant 4 of Figure 5 presents the citation map by sources. This analysis is important to identify which journals are more aligned with the topic of Industry 5.0, assisting researchers in sharing their research on the subject. The main sources by clusters are: Journal of Manufacturing Systems (green cluster), Sustainability (orange cluster), Applied Sciences (dark blue cluster), International Journal of Production Research (red cluster), Sensors (yellow cluster), and Journal of Industrial Integration and Management (light blue cluster).

The bibliometric analyses presented in this subsection will enable researchers in the field of Industry 5.0 to understand research in this area, evaluate its quality and impact, make informed decisions, and track its development over time. It will help identify trends and patterns in research on Industry 5.0, assess the quality and impact of specific research

on the subject, compare research across different countries and institutions, and aid in making more informed decisions (such as resource allocation for research and partnerships between research centers and institutions).

To conclude this section, it is important to mention that part of the bibliometric data, particularly the co-occurrence analysis conducted using VOSviewer®, was used for visualizing similarities among the collected data. It was combined with thematic analysis using Braun and Clarke's [12] approach, allowing for the identification, coding, and grouping of data into analysis categories, which will be discussed in the next section.

## 4. Discussion

### 4.1. Industry 5.0 from a Quadridimensional Perspective: Advancing through a New View of Neoindustrialization

A set of advanced technologies, represented by the Internet of Things (IoT), artificial intelligence (AI), and robotics, enable the creation of flexible and highly efficient production systems, characterizing Industry 4.0 [30]. Its adoption has been gaining momentum, positively impacting various sectors, and becoming the standard in the industry in recent years. However, these developments still fall short of achieving the desired outcomes [1] and have faced significant challenges. These challenges include the need to adapt organizational structures and traditional business models, as well as the need to develop solutions to ensure data security and interoperability between systems. Additionally, there is a concern that the environment is being neglected and that machines are being prioritized over humans [1,30].

In 2021, the European Commission formalized the call for a new era in the industry when it published the article titled "Industry 5.0: Towards a sustainable, human-centric and resilient European industry" This work, resulting from two workshops held in 2020, lays the groundwork for what is now known as Industry 5.0. It represents a new paradigm that embraces sustainability, human-centered approaches, and organizational resilience as its core values [1,4,5]. Therefore, Industry 5.0 fills the gap left by the previous paradigm by prioritizing social and environmental issues alongside technological innovation [5].

As per the vision outlined by the European Commission, Industry 5.0 is based on three interconnected key principles: the importance of the individual, sustainability, and resilience. This human-centered approach positions their needs and interests at the heart of the production process, shifting the focus from a technological view to a strategy oriented towards the individual and society [4,31]. In this scenario, technology is seen as a means, being used to adapt to the diverse needs of industry workers and consumers [31]. Moreover, it's imperative to create a work environment that is safe and inclusive, prioritizing physical and mental health, as well as well-being, and safeguarding the fundamental rights of workers, human dignity, and privacy. For the industry to respect the limits of our planet, it needs to be sustainable. This requires the development of circular processes that reuse and recycle natural resources, reduce waste and environmental impact, and drive a more efficient and effective circular economy. Organizational resilience, in turn, refers to the need to develop a more robust industrial production, able to protect against disruptions and ensure the supply and sustenance of critical infrastructures in times of crisis. The industry of the future must be resilient enough to quickly adjust to geopolitical changes and natural emergencies [4,5]. Industry 5.0 will allow the addressing of issues related to technology and human life not observed by Industry 4.0, in addition to providing greater scalability, flexibility, resilience, and efficiency in the industry [1,32].

Unlike the announcement made by the European Commission (2021), which indicates that Industry 5.0 should be human-centric, sustainable, and resilient, this study argues that this new approach to the industry should be envisioned from four perspectives of analysis. This additional perspective considers technology as an enabler of Industry 5.0, that is, a set of new technologies that will characterize this new industrial paradigm, fostering collaboration and interrelation between humans and machines [6,32]. In this new reality, where factories and devices are intelligent, humans work alongside machines

and connect to them through smart devices [33]. Production undergoes a transformation, making way for mass personalization in the industry. In this context, robots will have significant importance, as advancements in artificial intelligence allow them to be connected to the human mind through brain interfaces [34] functioning as collaborators rather than competitors [13]. The advent of this new industrial revolution will drive the advancement of interfaces between humans and machines, utilizing artificial intelligence (AI) algorithms. This evolution will enable enhanced integration, allowing for more efficient and faster automation while harnessing the full potential of the human brain [17]. This also means that robots will not take over factories anytime soon, alleviating concerns from the era of Industry 4.0 [18]. The transition from Industry 4.0 to Industry 5.0 means combining the best features of humans and machines, which will result in increased productivity [3,35].

For proponents of this approach, the main characteristic of Industry 5.0 is "Personalization", which can be described as "the design and production through various sensor data that will be directly linked, providing personalized products to users in real time" [3], p. 1, thus representing a neoindustrialization. In other words, this new reality allows customers to obtain products and services according to specific requirements, following a manufacturing process through the use of technologies that enable "design freedom", making products more personalized and enhancing manufacturing capabilities by utilizing programmable machines and intelligent sensor networks [15,18,36]. In this context of a new revolution, the industry meets the individual demands of customers, made possible by a paradigm shift in manufacturing, transitioning from customization to mass personalization [18].

Despite the conceptual differences, there is a consensus that Industry 5.0 envisions an industry that goes beyond the goals of efficiency and productivity, emphasizing the role and impact of the industry on society [4].

It is essential to emphasize this point, as it justifies the need for a new paradigmatic approach based on neoindustrialization to address the challenges imposed on organizations, governments, and society as a whole. These challenges, such as climate change, rapid consumption of non-renewable resources and energy, environmental pollution, and social injustice, among others, largely impact the world today and have gradually gained prominence, particularly in the past two hundred years, which coincides with the industrial revolutions.

### 4.2. A New Stage for the Industry: A Revolution or Just Another Number?

The numeral that follows the term "industry" indicates a paradigmatic evolution known as an industrial revolution. What determines this evolution? In other words, what characterizes a new industrial revolution? Can Industry 5.0 be seen as a new revolution in the industry? Looking back at the past can help answer these questions, so it is important to establish a temporal demarcation of the evolutionary stages of the industry from the perspective of neoindustrialization.

Over the past two centuries, four industrial revolutions have occurred, "each producing a higher level of technology" [1], p. 1. The first industrial revolution, known as Industry 1.0, originated around 1760 through the generation of mechanical power from water, steam, and fossil fuels [1,13]. The steam engine allowed the transition from an agrarian and feudal society to a new manufacturing process. This transition included the use of coal as the primary source of energy while trains became the main mode of transportation. The textile and steel industries dominated in terms of employment, production value, and invested capital. It was followed by the second industrial revolution, Industry 2.0, which took place in the first half of the 19th century, around the 1840s. It was enabled by electric power and the invention of the internal combustion engine, leading to a period of rapid industrialization using oil and electricity, facilitating the advent of assembly line factories and enabling mass production [1,13].

In the 20th century, around the 1960s and 1970s, the era of computers, transistors, and later on, silicon chips, electronic devices, and information technology (IT) emerged,

marking the Third Industrial Revolution (Industry 3.0) and introducing the industry to the concept of automation.

On the other hand, Industry 4.0 is an initiative by the German government that suggests the creation of smart factories with a simple and similar objective to previous revolutions: increasing productivity and achieving mass production with innovative technology [33]. It utilizes technologies such as artificial intelligence (AI), cloud computing, and the Internet of Things (IoT), enabling a real-time interface between the physical and virtual worlds (cyber-physical systems) [1,13]. Industry 4.0 is characterized by the integration of intelligent manufacturing systems and processes with advanced information and manufacturing technologies, enabling a flexible, intelligent, and reconfigurable production process [37].

In the technological search revolution of hyper-accumulation [38], evolutionary models of industrial dynamics based on the concepts of Industry 4.0, despite enabling several technological advancements, have limitations from an anthropocentric perspective in the industrial production process.

Industry 5.0 emerges as a complement to Industry 4.0, addressing the aspects that it does not fully cover, particularly issues related to social justice and sustainability. While still considering productivity and efficiency, Industry 5.0 brings the human-machine relationship to the forefront, placing the individual at the center of the production process [1,4,5].

The foundation for the construction of the principles of Industry 5.0 stemmed from society's and governments' concerns regarding the preference for machines over humans in the industry, along with the need to consider sustainable development and recognize the crucial role of humans in shaping the future of industrial development [28]. Moreover, even in conceptual terms, the excessive emphasis on technology and automation in Industry 4.0 distances it from the necessary paradigmatic alignment to serve as a solution to the challenges listed in the introduction of this paper [6]. This position has directed the search for a new paradigm that makes the industry (representing other organizations) more sustainable, human-centric, and resilient [4].

It is possible to identify common points regarding the industrial revolutions. A technological advent: steam power (Industry 1.0), electricity (Industry 2.0), computers (Industry 3.0), cyber-physical systems, and the Industrial Internet of Things (Industry 4.0). A specific phenomenon impacting the economy and society: the adoption of new agricultural techniques, leading to increased agricultural productivity and trade expansion (Industry 1.0); World War I, the Great Depression of 1929 (Industry 2.0); World War II (Industry 3.0); population aging, urbanization, growing demand for sustainability, and concerns about cybersecurity (Industry 4.0). According to [5], industrial revolutions are driven by transformative technological advancements that lead to fundamental changes in how the industry operates, with economic and social consequences. In other words, an industrial revolution is determined by a series of significant changes in the way production is carried out, accompanied by technological advancements, increased productivity, and changes in production methods and work organization. The transition from one industrial revolution to another is a complex and multifaceted process involving technological, economic, social, and political changes.

In this regard, the following question arises: are we already experiencing a new industrial revolution? What led to this change in the industry's approach? It is possible to identify several trends and changes that characterize the transition to the next phase of industrial evolution, such as: (a) advancements in disruptive technologies: quantum computing, biotechnology, nanotechnology, energy storage technology, among others; (b) integration of systems and technologies: the fifth industrial revolution will enable the creation of even more complex and interconnected systems, leading to greater efficiency and personalization in all aspects of production and services; (c) growing importance of sustainability: society increasingly recognizes the need to find solutions to environmental and social challenges. This new reality in the industry will allow for the development of technologies and processes that reduce environmental impact and improve quality of

life; (d) new forms of work and organization: the gig economy and remote work, as well as new business models and cooperation, will lead to greater flexibility and diversity in the workplace, as well as new forms of collaboration and innovation; (e) regulation and public policies: Regulation and public policies can play an important role in determining the direction and pace of the fifth industrial revolution, including innovation incentives, investments in infrastructure, data security, and privacy standards, among others.

The absence of a technological advent fuels critics of Industry 5.0, who do not see it as a new paradigm for the industry. In this case, it can be observed that the technologies indicated as enablers of Industry 5.0 are similar to those of the previous paradigm. On the other hand, as seen in previous revolutions, phenomena can occur that impact markets and the global economy, demanding new approaches and differentiated actions from the industry (e.g., the two world wars, the 1929 crisis, the COVID-19 pandemic, which impacted the global supply chain).

Thus, drawing a parallel between Industry 4.0 and 5.0, it can be inferred that the former is technology-driven, while Industry 5.0 is propelled by values, with a focus on issues such as sustainability and social responsibility [5], configuring itself as a neoindustrialization.

### 4.3. Neoindustrialization and the Resurgence of Industry's Protagonism: The Context of the Brazilian Industry

Industries are organizations that aim to transform raw materials into marketable goods and services. The industry originated with the industrial revolution and has evolved as mentioned in the previous subsection. From an economic point of view, the industry can be classified into four sectors: primary, secondary, tertiary, and quaternary. Industries in the primary sector are involved in raw material extraction. The secondary sector comprises industries engaged in processing, manufacturing, and construction using the raw materials extracted from the primary sector. The tertiary sector consists of service industries such as retailers, entertainment companies, and financial organizations. The quaternary sector encompasses innovative industries that use information and technology to improve processes and services, leading to societal improvements [39].

The Brazilian industrial sector is one of the largest job creators in Brazil and strengthens the entire productive sector. Research from the National Confederation of Industry (CNI) reveals that the sector employs 9.7 million Brazilians and accounts for 20.4% of the country's formal employment. The participation of the Brazilian industry in the global industry represents 1.5%, with Brazil ranking 13th among the most industrialized countries. However, Figure 6 indicates that these numbers have been experiencing a downward trend. Importantly, it is also worth mentioning that these numbers are prior to the COVID-19 pandemic, which had a significant impact on the productive sector and disrupted the global supply chain.

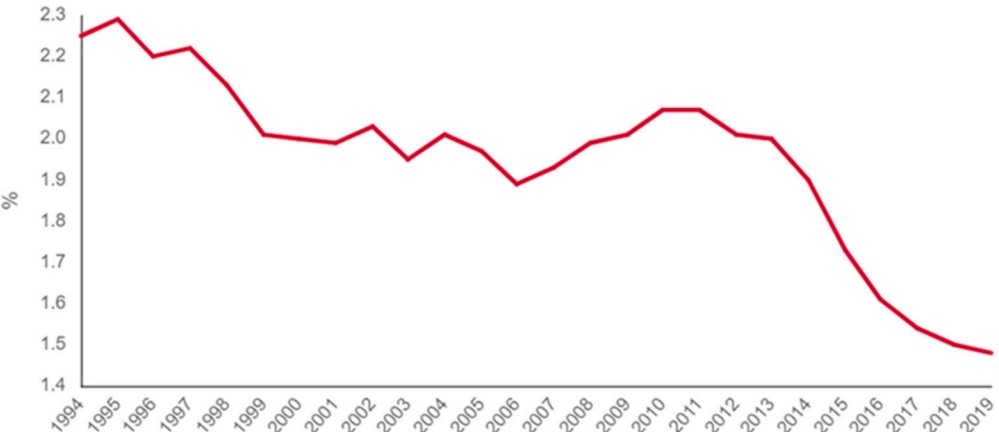

**Figure 6.** Participation of the Brazilian industry in the global industry [39].

The participation of the manufacturing industry in the gross domestic product (GDP) can be evaluated in two distinct periods, between the years 1947 and 2018, as observed in the graph in Figure 7.

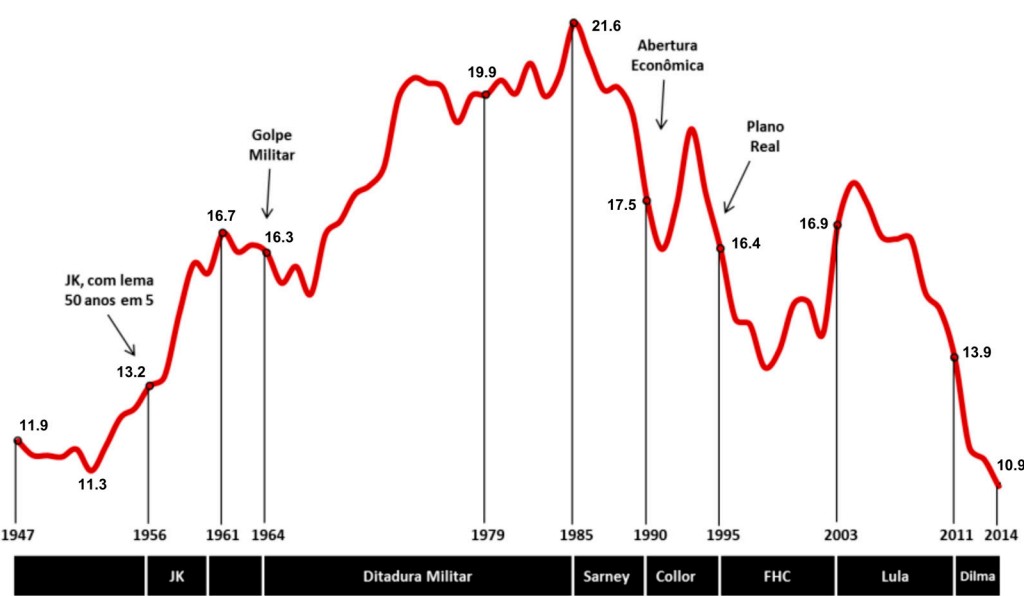

**Figure 7.** Participation of the Manufacturing Industry in the Brazilian GDP [40].

From the 1950s until 1985, the first period occurred, characterized by intense growth, diversification, and consolidation of the Brazilian industrial structure. During this period, the participation of the manufacturing industry in the GDP almost doubled, rising from 11.3% in 1952 to 21.6% in 1985. The second period, which began in 1986, is characterized by a significant loss of industry's share in the country's aggregate production [40].

The peak of the manufacturing industry in Brazil occurred in 1985, and since then, the industry's importance has been gradually declining, with services gaining in significance [41]. According to data from the Brazilian Institute of Geography and Statistics (IBGE) and estimates from FIESP [40], the share of the Manufacturing Industry in the GDP declined by approximately 10 percentage points over the last period, reaching 10.9% in 2018. This phenomenon is known as deindustrialization [40,41]. This scenario is characterized by the decrease in employment opportunities in the industrial sector and its contribution to the composition of the GDP, caused by the loss of competitiveness and relative technological backwardness, the absence of public policies for the industrial sector, and an unfavorable economic environment, associated with dependence on foreign capital and technology.

Furthermore, a study conducted by the Federation of Industries of Rio de Janeiro [42] states that Brazilian industry is in transition between the second and third industrial revolutions [43]. A study conducted by the National Confederation of Industry [44], cited by [43], revealed that 43% of the 2225 surveyed companies were unaware of the technologies that could drive competitiveness in Industry 4.0. Furthermore, the lack of knowledge was more pronounced among small companies (57%) compared to large companies (32%). Additionally, 48% of the companies were using at least one new technology, with a focus on digital industrial automation (27%), collection and processing of large amounts of data (9%), operation of flexible autonomous lines (8%), cloud services (6%), and the Internet of Things (4%). According to the study, challenges for Industry 4.0 included implementation costs, lack of clarity in return on investment, organizational structure, national infrastructure, employee qualifications, and industrial policies/regulations [43].

Discussing Industry 5.0 after reading the previous paragraphs may seem presumptuous and ambitious, as it appears to be a distant scenario for Brazil. However, it is important to emphasize that the Brazilian industrial context is represented by diverse realities. For instance, the seven states comprising the South and Southeast regions of Brazil account

for 73.9% of the country's industrial GDP, despite Brazil having 27 federative units. This panorama not only demonstrates the disparity in the industry across Brazilian regions but also highlights the need for the Brazilian industry to reclaim its prominent role of decades past. To achieve this, a process of Neoindustrialization is necessary, which involves restructuring the industrial sector through the incorporation of new technologies, increased productivity, and a focus on higher value-added industries. This transition will entail shifting from traditional industries to advanced industries such as information technology, biotechnology, and renewable energy, among others. These actions will serve as a strategic response to contemporary economic challenges, including international competition, process automation, and the growing demand for sustainable and environmentally responsible solutions, aligning with the advent of Industry 5.0.

To achieve this, it is necessary to create an environment conducive to innovation and entrepreneurship that promotes the development of qualified human capital, investments in research and development, and strengthens partnerships between the public and private sectors. A successful neoindustrialization policy will generate high-quality jobs and foster economic growth.

In order for the process of neoindustrialization to be effective and successful, the following critical factors, as presented in Table 2, need to be taken into account:

**Table 2.** Critical Success Factors for the Implementation of Neoindustrialization.

| Critical Success Factor | Description |
|---|---|
| Technological Infrastructure | - Improvement in connectivity and access to high-speed internet<br>- Implementation of advanced communication networks (e.g., 5G, 6G) |
| Qualified Human Capital | - Investments in education and professional training focused on digital and technological skills<br>- Requalification programs for workers transitioning from the traditional industrial economy to the digital economy |
| Innovation and Technological Development | - Encouragement of research and development through partnerships between universities, research institutes, and industry<br>- Creation of incentives for the development of new technologies and digital solutions |
| Favorable Business Environment | - Simplification of processes and regulations to facilitate entrepreneurship and attract investments<br>- Fiscal and financial incentives for companies adopting digital technologies and sustainable practices |
| Sustainability and Social Responsibility | - Promotion of sustainable and environmentally responsible practices in industrial production<br>- Encouragement of social inclusion and generation of quality jobs in the industrial sector. |

The term Neoindustrialization is more suitable to describe this new phase of industrialization, in which companies in the industrial sector will need to combine technology with new strategies and organizational models, as well as undertake organizational changes that align their structure, operations, human resources, and new practices to meet the demands of society and all stakeholders involved in this process.

### 4.4. The Enabling Technologies of Industry 5.0: From Technocentrism to Mass Personalization

One of the perspectives highlighted in Section 4.1 indicates that technologies are also crucial in Industry 5.0. They enable the interaction between humans and robots, bringing humans back to the factory floor and presenting opportunities that position this new Industrial Revolution as the future of the industry.

Several technological trends are empowering the industry to increase production and deliver personalized products to their customers, while also enabling interaction between

humans and machines [14]. Table 3 summarizes a range of technologies that are part of Industry 5.0.

**Table 3.** List of Enabling Technologies of Industry 5.0. Source: [14,18].

| Technologies | Definition |
|---|---|
| Edge Computing | Edge Computing (EC) is a data processing approach at the network edge that offers benefits such as low latency, increased energy efficiency, and enhanced security. Industries can access local data and minimize the volume sent to centralized servers, enabling proactive analytics and smarter decision-making. |
| Digital Twins | Digital Twins (DT) are digital replicas that enable mass customization, and the seamless flow of data between the physical, digital, and cyber space is necessary for their application. In Industry 5.0, DT enables analysis, monitoring, and prevention of issues before they occur in the real world, offering significant value for the development of personalized products and innovative business models. With IoT and advancements in AI, ML, and big data analytics, DT reduces maintenance costs, improves system performance, and helps prevent major financial losses. |
| Collaborative Robots | Cobots are robots designed to work alongside humans, providing increased efficiency and safety in the work environment. They have the ability to detect unforeseen impacts and immediately stop when they detect objects in their path. While efficient in large-scale production, managing human connections remains important in tasks that require critical thinking and customization. |
| Internet of Everything | Internet of Everything (IoE) connects people, processes, information, and objects, offering benefits for Industry 5.0 such as enhancing customer experience and reducing operational costs. IoE can optimize the supply chain, reduce waste, and improve production processes. Wireless technology and sensors are used for information exchange, such as in the Internet of Medical Things. |
| Big Data | Big Data is a technology that stores large amounts of complex data using IoT devices and provides significant services to manufacturers and service providers. Big Data Analytics enables the analysis of large volumes of data, allowing for mass customization and a better understanding of consumer behavior in Industry 5.0. With the integration of Big Data and IoT, real-time information can be collected to optimize production, reduce costs, and make more informed decisions. |
| Blockchain | Blockchain is a secure and decentralized technology that protects customer data against deletion, tampering, and revision, making it suitable for handling data privacy and traceability. It can be used to create distributed management platforms, providing transparency and immutability for significant event records in Industry 5.0. Additionally, it enables the execution of smart contracts to enforce security measures and automate processes. |
| 6G | The 6G technology can offer valuable services for Industry 5.0, with dense infrastructure, reduced latency, and integrated AI capabilities. 6G networks can enhance the performance of Industry 5.0 applications, but energy efficiency needs to be ensured. Quantum and free-space optical communication can help address high data rate challenges. |
| Artificial Intelligence | Artificial intelligence provides human-like capabilities to perform tasks in the manufacturing field, enabling the solution of complex problems in a faster and more cost-effective manner. Additionally, AI is capable of understanding the functioning of the human brain and efficiently executing high-level tasks, enhancing the thinking process and directing the system towards powerful and predictive intelligence. |

With greater manufacturing flexibility, this neoindustrialization can meet and even exceed customer satisfaction requirements, personalization, enhanced productivity, efficiency, and product quality. Raw material waste is reduced as products are manufactured according to customer demand. Industry 5.0 utilizes devices, systems, automation, and intelligent and innovative materials [18].

In addition to the mentioned technologies, software, applications, and platforms play a crucial role in enhancing experiences and implementing intelligent manufacturing, from the perspective of Industry 5.0. These technologies enable control and monitoring of manufacturing processes, facilitating efficient and high-quality production. Regarding the design and manufacturing of complex-shaped products, the mentioned technologies are essential. They allow for 3D modeling and simulation of products before physical production, reducing the need for storing large quantities of finished products. Instead, designs can be stored digitally and produced on-demand, reducing inventory costs and

minimizing waste [18]. To make the above a reality, technology will have a crucial role in operationalizing Industry 5.0.

*4.5. The Human-Machine Relationship in the Industry: A Path to Be Explored*

The relationship between humans and machines is referred to in the literature as Human-Machine Interaction (HMI) and can be characterized as "a form of communication and interaction between human users and machines in a dynamic environment through various interfaces" [45], p. 23. Throughout successive industrial revolutions and changes in manufacturing paradigms, this relationship has undergone significant transformations, influenced by human needs and the technologies available in each period [46].

As technology advances, machines acquire new functions, capabilities, and even skills that were once exclusive to humans, such as vision, inference, and classification [46]. This interaction has been present since humans began to build tools [45]. With the emergence of new revolutions and changes in manufacturing paradigms, the relationship between humans and machines becomes increasingly relevant and complex [46].

The next industrial evolution is characterized by leveraging the creativity and innovative potential of human experts in collaboration with efficient, intelligent, and precise machines in order to achieve resource efficiency and user/consumer-driven custom manufacturing solutions [14,47,48].

Humans have a limit to their cognitive capacity, which can affect their performance in certain tasks. On the other hand, machines have the ability to process massive amounts of information. By utilizing machines, which have greater processing capacity, it is possible to reduce the cognitive load on humans. This allows humans to focus on activities that require uniquely human skills, while machines handle the task of information processing. This task distribution aims to optimize overall performance and improve efficiency by leveraging the complementary abilities of humans and machines [49]. The adoption of Industry 5.0 technologies, therefore, will not undermine human value; on the contrary, it will enable the integration of human intelligence and machine intelligence in a collaborative environment [3].

In this context, dedicated manufacturing systems are being replaced by intelligent and flexible manufacturing systems with the introduction of collaborative machines driven by artificial intelligence, transforming the way work is performed on the factory floor [31]. The focus shifts to the role of humans in this technological transition, with attention to the collaboration between humans and robots in Industry 5.0 [48], aiming to enhance human capabilities and well-being [31].

To achieve this, it is necessary to develop a human-centered manufacturing system that optimizes its benefits for humans. Based on this premise, Lu et al. [31] propose a framework that seeks the symbiosis between humans and machines, where both form intelligent teams to collectively perceive, reason, and act in response to manufacturing tasks and contingencies.

The human-centric approach focuses on satisfying needs and promoting the well-being of individuals, taking into consideration their capacity to learn and adapt. These characteristics well define the new concept of neoindustrialization, highlighting the importance of placing humans at the center of the industrial processes and considering their well-being and adaptability as essential factors.

By adopting this approach, as artificial intelligence technologies advance, Industry 5.0 will transform production systems worldwide, allowing repetitive tasks to be performed by machines, freeing human workers for more creative and challenging activities [47]. Empathetic machines and high-performing human colleagues in dynamically coexisting environments will make manufacturing more resilient, flexible, and sustainable [31].

The collaborative relationship between humans and machines, referred to as human-machine symbiosis by [31], can enable the following benefits: (a) human well-being: reduction of stressful and repetitive operations that can expose the worker to potential health and safety risks; (b) flexibility in manufacturing: manufacturing systems and processes can be

instantly reconfigured to respond to product dynamics, human behavior, and production systems; (c) development of human and machine capability: with intelligent AI-based algorithms, humans and machines can learn and develop their capabilities through joint work experiences; (d) increased efficiency and productivity: machines can help automate repetitive tasks and production processes, allowing workers to focus on more complex tasks that require unique human skills; (e) improved product quality: machines can perform tasks with greater precision and consistency than humans, which can lead to an improvement in the quality of the final product; (f) enhanced workplace safety: machines can handle hazardous or repetitive tasks that pose risks to worker safety, reducing the risk of injuries or workplace accidents; (g) higher worker satisfaction: by automating repetitive tasks, workers can focus on tasks that require human skills such as creativity and critical thinking, making the work more challenging and interesting; (h) cost reduction: automation can reduce the time and cost associated with production, as well as reduce material waste; (i) better decision-making: machines can provide real-time data analytics and insights, enabling workers to make more informed and effective decisions [31,48,50]

Despite the technological advancements and digitalization driving Industry 5.0, there are still challenges to be faced, including: (a) lack of skills and training: the introduction of advanced technology will require skills and training that workers currently lack. In this regard, according to [21], the future workforce should be prepared to discern and understand the different production systems in order to make relevant decisions regarding the various approaches to work: whether through exclusive human effort, exclusive technological effort, or a harmonious collaboration between both. It is essential for professionals to have the experience and knowledge required to make informed choices, aiming to optimize production processes and achieve superior results. The lack of proper skills and training can lead to errors and reduce overall production efficiency; (b) resistance to change: some workers may resist the adoption of new technologies, concerned about job loss or changes in their roles. This can create a climate of mistrust and resistance that hinders the adoption of new technologies; (c) privacy and security concerns: the introduction of advanced technology can increase the risk of privacy breaches and data security, especially if sensitive worker information is collected and stored by machines; (d) liability in case of failures: Automation may reduce the need for workers in certain tasks, but it can also increase liability in case of failures or errors made by the machines; (e) high cost: the introduction of advanced technology can be expensive and may require significant investment in hardware and software, as well as ongoing updates and maintenance; (f) excessive dependency: excessive reliance on advanced technology can make production vulnerable to technical failures or disruptions in power supply or connectivity [31,48,50]. Table 4 summarizes the benefits, advantages, and opportunities, as well as the challenges and threats of human-machine interaction.

**Table 4.** Challenges/threats and benefits/advantages/opportunities of human-machine interaction.

| Benefits/Advantages/Opportunities | Challenges/Threats |
|---|---|
| Human well-being | Lack of skills and training |
| Manufacturing flexibility | Resistance to change |
| Development of human and machine capabilities | Privacy and security concerns |
| Increased efficiency and productivity | Accountability in case of failures |
| Improved product quality | High cost |
| Improved workplace safety | Excessive dependence |
| Higher worker satisfaction | |
| Cost reduction | |
| Improved decision-making | |

The challenges/threats and benefits/advantages/opportunities of human-machine interaction are summarized in the table below:

In the context of the neoindustrialization of Industry 5.0, the human-machine interaction represents one of its key characteristics and challenges. This new stage of industrial evolution is not just about automation and digitalization, but rather the harmonious and collaborative integration between humans and intelligent systems. It seeks to combine the best of both worlds: the unique capacity of humans for creativity, complex problem-solving, and understanding social and emotional nuances, with the efficiency, precision, and data processing capability of machines.

### *4.6. Study Limitations and Future Research Agenda*

During the study, some limitations were identified that should be considered when analyzing the obtained results. These limitations include: (a) the choice to portray the context of the Brazilian industry over other realities such as the European or Latin American industries; (b) another limitation to be considered is that, although the objective of this study was to identify the current state of research on Industry 5.0, a thorough analysis of each aspect of the presented analytical categories was not conducted, demanding further depth in subsequent literature reviews, such as an integrative or systematic review; (c) furthermore, despite detailing all the methodological procedures, selection biases may have occurred in the choice of research sources and articles included in this review; (d) finally, it is necessary to acknowledge the linguistic limitation of this study, as only articles in English and Portuguese were considered, potentially excluding relevant research in other languages.

Considering these limitations, it is possible to highlight a future research agenda to advance the understanding of Industry 5.0. It is recommended that future studies address the following topics: (a) conducting theoretical/empirical studies to assess the maturity level of the Brazilian industry compared to other global industry realities; (b) conducting in-depth studies on each of the categories presented in this review, through systematic reviews or empirical research; (c) investigating the implementation of Industry 5.0-related initiatives in detail, to understand the challenges faced, best practices adopted, and results achieved; (d) exploring the applications of emerging technologies such as artificial intelligence, Internet of Things, virtual and augmented reality, and cloud computing in the context of Industry 5.0; (e) examining the social and ethical implications of human-machine interaction in Industry 5.0, including issues of privacy, security, responsibility, and the impact on the job market; (f) considering the analysis of policies and regulations necessary to promote responsible adoption of Industry 5.0, including aspects such as data protection and technology governance.

By addressing these issues in future studies, it will be possible to advance the understanding and development of Industry 5.0, contributing to the progress of the industrial sector and society as a whole.

### 5. Conclusions

The process of global urbanization was intensified by the various stages of industrialization, which triggered many of the problems mentioned in this study (for example, depletion of natural resources, global warming, increasing economic disparity, among others). These problems arose or were exacerbated, largely due to the harmful way in which industrial development occurred. As a way to reverse this situation of unsustainability and cyber vulnerability, a new stage for the industry arises from the observation that Industry 4.0 prioritizes technology at the expense of human beings and the environment.

The green and digital perspective advocated by the European Union is already emerging as a trend in academic debates, and it is up to academia to bridge the gap, demonstrating its importance and seeking the adoption of this new paradigm in the industry, of neoindustrialization.

Industry 5.0 emerges as the alternative for the long-term sustainability of industry by emphasizing the role of research and innovation, aligning production with respect for the planet's limits, and positioning the industrial worker at the heart of the production process. As advocated by the European Commission, and expanded upon in this study, Industry 5.0 covers four dimensions that are representative of its fundamental principles, portraying an industry that broadens its objectives beyond just efficiency and productivity, emphasizes its societal role and contribution, while shifting its focus towards a sustainable, human-centered, resilient industrial transition and the harmonious and collaborative integration of humans and intelligent systems.

On the other hand, this study understands that Industry 5.0 can be viewed from an additional perspective that considers the interaction between humans and machines, enabled by technology, allowing for additional features in this new approach to the industry, particularly mass customization of products and services, enabling a unique and innovative experience for consumers and generating value and competitive advantage for industrial organizations, emerging as a neoindustrialization.

The industry should consider this opportunity, and this work should be seen as a call to embrace this new reality, which anticipates future trends and highlights the need for changes. Those who embrace this trend will be at the forefront of the next stage of the Industrial Revolution.

The COVID-19 crisis has indicated the need to rethink existing work approaches, as well as highlighting the vulnerability of global supply chains. Therefore, industries need to be better prepared for the future, aiming to be resilient, sustainable, and integrate human values with technology in order to achieve sustainable development goals. Moreover, before being a resilient provider of prosperity, meeting society's demands and respecting planetary boundaries, the industry must balance this agenda with the need to pursue competitiveness and productivity, considering the technological advancements characteristic of a new industrial revolution.

The industry needs to regain its prominence in the economic landscape, and for that, it needs to adopt a process of Neoindustrialization, a term that describes this new phase of industrialization, in which companies in the industrial sector will need to combine technology with new strategies and organizational models, as well as undertake organizational changes that align their structure, operations, human resources, and practices with the demands of society and all stakeholders involved in this process.

To achieve this, it is necessary to create an environment conducive to innovation and entrepreneurship, promoting the development of qualified human capital, investments in research and development, and strengthening partnerships between the public and private sectors. A successful neoindustrialization policy will generate quality jobs and foster economic growth.

In light of all the above, it is possible to conclude that Industry 5.0 is the prevailing paradigm in the 21st-century industry. It is not a matter of speculation; it is an inseparable and inevitable reality. Otherwise, the industry will be relegated to a secondary role in the process of digital and social transformation.

**Author Contributions:** Conceptualization, R.P. and N.d.S.; methodology, R.P. and N.d.S.; formal analysis, R.P. and N.d.S.; investigation, R.P. and N.d.S.; writing—original draft preparation, R.P. and N.d.S.; writing—review and editing, R.P. and N.d.S.; supervision, N.d.S. All authors have read and agreed to the published version of the manuscript.

**Funding:** This research received no external funding.

**Data Availability Statement:** Not applicable.

**Conflicts of Interest:** The authors declare no conflict of interest.

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
