# Peer review of "Neoindustrialization—Reflections on a New Paradigmatic Approach for the Industry: A Scoping Review on Industry 5.0"

_logistics_

Round 1
Reviewer 1 Report
Authors presented interested and actual topic about Industry 5.0. The quality of figures should be improved because now they are illegible and some of them dont look profesionnally and adequate for paper (such as figure 4-figure 9). It would be nice also to mention about Industry 5.0 transformative vision for Europe. And subchapter 4.5 seems to be to short and doen't contain summary (maybe in the form of tables) for pros and cons in terms of human machine relationships. Please specify research goals for the entire article not only theoretical review of papers from databases.
Im not qualified to comment quality of English
Author Response
Dear Reviewer,
We would like to express our gratitude for the valuable comments and suggestions provided on our article entitled "Neoindustrialization - Reflections on a new paradigmatic approach for the industry: a scoping review on Industry 5.0." We sincerely appreciate the time and effort you have dedicated to reviewing our work. Below, we address each of the points in detail, highlighting the revisions and improvements made in response to each of them.
Point 1: "The quality of figures should be improved because now they are illegible and some of them don't look professionally and adequate for paper (such as figure 4-figure 9)" Response: We appreciate your comment. We have reviewed and improved the quality of the figures. Regarding the adequacy of figures 4 and 9, we would like to clarify that the first one is an image generated by the Vosviewer software, which allows for occurrence analysis and visualization of similarities, an essential step for the bibliometric analysis of the study. As for figure 9, it was developed to represent the main entity of the Brazilian industrial sector, FIESP, and it appropriately illustrates and contextualizes the chronological evolution of the Brazilian industry.
Point 2: "It would be nice also to mention about Industry 5.0 transformative vision for Europe" Response: Thank you for the observation. At this moment, we find it more appropriate to focus on the reality of the Brazilian industry. In future studies with an empirical focus, it would be valuable to conduct a comparative analysis between the Brazilian and European industry contexts.
Point 3: "Subchapter 4.5 seems to be too short and doesn't contain a summary (maybe in the form of tables) for pros and cons in terms of human-machine relationships." Response: We appreciate the suggestion. Subsection 4.5 has been modified and expanded to include a table that covers the positive and negative aspects of human-machine interaction in the context of Industry 5.0.
Point 4: "Please specify research goals for the entire article not only theoretical review of papers from databases." Response: Thank you for your feedback. The objective of this work, being a scoping review, is to identify the current state of knowledge in the area under review. However, the objective has been reformulated to address the suggestion.
Once again, we appreciate the thorough review of our article and the constructive observations. We believe that the changes implemented in response to your points have substantially strengthened the work. We hope that the revised manuscript meets the expectations of the journal and the reviewers.
Reviewer 2 Report
This is a very good paper giving a literature review in the field of Industry 5.0 as the new industrial paradigm. I would advise authors to provide some minor revisions which are following:
1. Please explain why the literature was specially searched through the the main business and management journals in Brazil as well as the proceedings of EnANPAD. I am sure this will give an authors directions for the future research, but it would be good to reflect at the current state of the Brazilian or Latin America industry for reader to have an insight about how is it now and compare to the future state defined by Industry 5.0
2. In conclusion section, please add the possible pathways of the future research.
Author Response
Dear Reviewer,
We would like to express our gratitude for the valuable comments and suggestions provided on our article entitled "Neoindustrialization - Reflections on a new paradigmatic approach for the industry: a scoping review on Industry 5.0." We sincerely appreciate the time and effort you have dedicated to reviewing our work. Below, we address each of the points in detail, highlighting the revisions and improvements made in response to each of them.
Point 1: "Please explain why the literature was specially searched through the main business and management journals in Brazil as well as the proceedings of EnANPAD" Response: We appreciate your comment. The inclusion of the main business and management journals in Brazil, as well as the proceedings of EnANPAD, as sources for the study aims to reflect the current state of research development in Brazil on Industry 5.0, which has been shown to be in its early or incipient stage, lacking studies like the one undertaken by this article.
Point 2: "I am sure this will give authors directions for future research, but it would be good to reflect on the current state of the Brazilian or Latin American industry for readers to have an insight about how it is now and compare to the future state defined by Industry 5.0" Response: Thank you for the observation. We have revised subsection 4.3 to include further reflections on the current state of the Brazilian industry and propose neoindustrialization as a solution to this scenario.
Point 3: "In the conclusion section, please add the possible pathways of future research." Response: We appreciate the suggestion. We have created subsection 4.6, which discusses the limitations of the study and presents a future research agenda.
Once again, we appreciate the thorough review of our article and the constructive observations. We believe that the changes implemented in response to your points have substantially strengthened the work. We hope that the revised manuscript meets the expectations of the journal and the reviewers.